# Quality Improvement in a Pediatric Echocardiography Laboratory: A Collaborative Process

**DOI:** 10.3390/children9121845

**Published:** 2022-11-28

**Authors:** Danielle Sganga, Sarina Behera, Meaghan J. Beattie, Katie Jo Stauffer, Aubrey Burlinson, Leo Lopez, Elif Seda Selamet Tierney

**Affiliations:** 1Department of Cardiology, Boston Children’s Hospital, Harvard Medical School, Boston, MA 02115, USA; 2Division of Pediatric Cardiology, Department of Pediatrics, Lucile Packard Children’s Hospital, Stanford University School of Medicine, Palo Alto, CA 94304, USA

**Keywords:** quality improvement, diagnostic errors, pediatric echocardiography laboratory, congenital heart disease

## Abstract

Transthoracic echocardiography (TTE) is an essential tool for diagnosis and management of congenital heart disease. Pediatric echocardiography presents unique challenges including complex anatomy, variable patient cooperation and provider expertise. Diagnostic errors inevitably occur. We designed a collaborative and stepwise quality improvement (QI) process to address diagnostic errors within our laboratory. We retrospectively reviewed medical records to identify diagnostic TTE errors in 100 consecutive cardiac surgery patients ≤ 5 years old (July 2020–January 2021). We identified 18 diagnostic errors. Most errors had minor impact (14/18), and 13 were preventable or possibly preventable. We presented these results to our sonographers and faculty and requested input on preventing and managing diagnostic errors. Our root cause analysis based on their responses yielded 7 areas for improvement (imaging, reporting, systems, time, environment, people, QI processes). Our faculty and sonographers chose QI processes and imaging as initial areas for intervention. We defined our SMART goal as a 10% reduction in diagnostic errors. We implemented interventions focused on QI processes. On initial follow up in May 2022, we identified 7 errors in 70 patients (44% reduction in error rate). Utilizing a stepwise and team-based approach, we successfully developed QI initiatives in our echocardiography laboratory. This approach can serve as a model for a collaborative QI process in other institutions.

## 1. Introduction

Pediatric and congenital heart disease is by nature heterogenous and complex. Transthoracic echocardiography (TTE) is essential for the diagnosis and guidance of clinical management and surgical and catheter-based interventions in these patients. TTE poses minimal risk for the pediatric patient compared to other imaging modalities, as it is non-invasive and does not involve radiation nor typically requires sedation. However, acquiring images in a pediatric patient population presents unique challenges, including the balance of variable patient cooperation with time management. Despite highly trained personnel and carefully developed imaging protocols, inaccuracies in imaging measurements and interpretation can occur. Quality improvement (QI) is a critical tool for identifying and addressing errors in a systematic and clinically meaningful way.

Key organizations, including the Intersocietal Accreditation Committee (IAC) and American Society of Echocardiography (ASE) have recommendations and requirements for routine and continuous QI reviews. The IAC recommends pediatric echocardiography laboratories perform quarterly evaluations of study completeness as well as technical and interpretive quality. They also recommend holding at least 2 laboratory QI meetings yearly [1]. ASE recommends the annual review of 5–10 studies performed by sonographers and physicians to quantify adherence to imaging protocols, the periodic review of qualitative and quantitative TTE interpretation for accuracy and intrareader reliability for every reader, as well as a continuous quality improvement plan consisting of case reviews with cross-modality comparisons [2]. The goals of these recommendations are to improve thoroughness and accuracy within a laboratory, but the process by which a laboratory achieves these goals is indeterminate. 

Routine review of preoperative echocardiograms, as described by the Adult Congenital and Pediatric Cardiology Section of the ACC [3], can be utilized to identify diagnostic errors of various impact. The echocardiography laboratory at Lucile Packard Children’s Hospital at Stanford performs approximately 13,000 echocardiograms annually that are vital to clinical care of children with congenital and acquired heart disease. In this study, we aimed to address diagnostic errors using a collaborative laboratory-wide process to develop and initiate quality improvement interventions (Figure 1). 

## 2. Materials and Methods

### 2.1. Diagnostic Error Identification and Classification

We identified a baseline TTE diagnostic error rate in 100 consecutive cardiac surgery patients at Lucile Packard Children’s Hospital at Stanford over 6 months (July 2020–January 2021). Patients were identified using our institution’s Society of Thoracic Surgeons database. Inclusion criteria were the completion of a full preoperative TTE and age < 5 years to limit the effect of poor echocardiographic windows. We excluded repeat instances of the same patient and any patient with a preoperative TTE described as limited or focused.

We performed a systematic retrospective review of the electronic medical record (EMR) to identify diagnostic TTE errors. We compared preoperative TEE reports with additional EMR findings, including operative reports, cardiac catheterization reports, and other pre- or postoperative imaging reports to identify discrepancies. We classified errors based on type, severity, and preventability, using previously published guidelines [4]. We also classified errors based on anatomic segment.

### 2.2. QI Methodology

We used our baseline diagnostic error rate to inform and develop a SMART (Specific, Measurable, Achievable, Relevant, Timely) goal. An electronic survey with baseline diagnostic error information was sent to all members of the echocardiography laboratory (faculty and sonographers), along with a request for submissions of ways to prevent and manage diagnostic errors. In reviewing these responses, we created a fishbone diagram, organizing the feedback into distinct categories representing potential key drivers of diagnostic errors.

Following this, we distributed a second electronic survey to the same group of faculty and sonographers, soliciting input on prioritizing areas of intervention. Our investigator team, which includes physician and sonographer representation, utilized survey responses to develop feasible QI interventions. We rolled out interventions in the echocardiography laboratory over the subsequent weeks, then reevaluated our diagnostic error rate.

## 3. Results

### 3.1. Baseline State

We identified 18 diagnostic errors in 17 patients by reviewing 100 consecutive surgical patient cases that met inclusion criteria (error rate of 18%). We analyzed these errors and categorized them based on anatomic segment, error type (false positive, false negative, discrepant diagnosis), impact severity, and preventability, as previously described in the literature (Table 1) [4]. Briefly, errors were considered minor if they had no impact on clinical course or patient management, moderate if they impacted patient management or led to an transient adverse event, and major if they resulted in an adverse event. Errors were considered possibly preventable if they could have been avoided with improved technique or imaging conditions, and preventable if they could have been avoided with the available images. Most errors had minor clinical impact (14/18) and 13 were considered preventable or possibly preventable.

### 3.2. Root Cause Analysis

With this baseline state information, we defined a quality improvement objective using SMART goal characteristics. SMART is an acronym that stands for specific, measurable, actionable, relevant, and timed. It is commonly used when creating QI project goals to help focus interventions and ensure any subsequent changes, positive or negative, are recognized and quantifiable. Our stated goal is to reduce diagnostic echocardiography errors in pre-operative patients 0–5 years of age by 10%, to an error rate ≤ 16.2%, over 4 months.

An initial anonymous survey sent out to all stakeholders in the echocardiography laboratory asked two key questions: (1) What are ways to prevent diagnostic errors in our laboratory? and (2) How should diagnostic errors be managed once identified? There was a 72% response rate, with 16 of 22 lab members participating in the survey.

The fishbone diagram served as a tool to organize possible root causes of diagnostic errors and suboptimal error management (Figure 2). Fishbone diagrams aid in the identification and analysis of multifactorial or multiple potential causes for a problem. We classified responses into 7 major areas: QI processes, imaging, reporting, time, personnel, environment, or systems/workflow.

### 3.3. Intervention Design and Implementation

The fishbone diagram was shared with the faculty and sonographers in the echocardiography laboratory. A second anonymous survey was created to prioritize interventions related to each area identified in the fishbone diagram. This survey had an 86% response rate. Responses are shown in a Pareto diagram (Figure 3), with the 2 most common answers being (1) imaging and (2) QI processes.

With these two areas to prioritize, our initial intervention focused on improving our QI review processes. Prior to this project, our laboratory held quarterly QI meetings for internal review of diagnostic errors. This practice continued. To enhance our QI review process, we implemented an error notification process for involved faculty and sonographers. We began recording meeting minutes for distribution to the team. We also created a database to keep track of all reported errors over time, to serve as both a method of tracking diagnostic error trends and a teaching resource. Finally, we added a section focused on “great catches” to our quarterly QI meeting agenda to celebrate correctly made challenging diagnoses. Additional QI process and imaging interventions continue to be developed and are ongoing.

### 3.4. Follow Up Evaluation

After the implementation of our initial QI interventions, we repeated a systematic EMR review to obtain a follow-up preoperative TTE diagnostic error rate. Inclusion criteria again were the completion of a full preoperative TTE and patient age ≤ 5 years. We conducted this systematic review over four months (January 2022–April 2022) as defined in our previously stated SMART goal. Seventy patients met inclusion criteria for review, and we identified seven diagnostic errors in six patients (Table 2). These errors were again analyzed and categorized based on anatomic segment, error type, severity, and preventability [4]. All seven errors were categorized as minor in terms of clinical impact.

## 4. Discussion

### 4.1. Discussion

Quality improvement is now widely regarded as an essential part of any echocardiography laboratory. Routine QI practices can help laboratories to retain a high-quality standard and to minimize diagnostic errors and their potential clinical impacts on patients. Excellent and thorough guidelines exist for pediatric and adult echocardiographic laboratories regarding the maintenance of facility and equipment standards, appropriate personnel training, and study completeness and imaging protocols [2,5,6].

Over the last decade, as the importance of QI work is increasingly recognized, there have been some relevant research efforts within pediatric echocardiography laboratories. Some QI efforts have focused on the utility of adherence to guidelines and imaging protocols [7,8,9,10]. Other pediatric cardiology QI efforts focus on the appropriate use of pediatric echocardiography in the clinical setting [11,12,13,14,15]. While all these efforts have improved the overall quality and accuracy of pediatric echocardiograms, it is inevitable that some diagnostic errors occur. The pediatric echocardiography laboratory, like any modern workplace, is dynamic by nature. Personnel, equipment, and technology continuously change. Academic medical centers train learners at many stages of their careers. Pediatric echocardiography laboratories will always need to incorporate new people, skills, and operations into their workflows. Thus, the ASE, IAC, and the Adult Congenital and Pediatric Cardiology Quality Network recommend continuous evaluation of echocardiographic imaging and interpretation in pediatric laboratories [1,2,3].

To further these efforts, it is important that we, as pediatric cardiologists and sonographers, find ways to effectively identify the causes of diagnostic errors and address them. Existing QI methodology is well suited to this task and readily applicable to the pediatric echocardiography laboratory. A key aspect of our QI process was the collaborative nature of diagnostic error review and initiative development. Utilizing anonymous surveys allowed us to increase echocardiography laboratory participation and ensure buy-in. The opportunities to develop QI initiatives are abundant in any laboratory, but our goal was to focus our initiatives in the areas that both cardiologists and sonographers felt were most important.

We recognize that different echocardiography laboratories will have different goals and definitions for quality improvement. Regardless of these differences, we believe the quality improvement process (as depicted in Figure 1) is a helpful tool that can be applied broadly to any laboratory. Once errors are identified and reviewed, root cause analysis can help identify overarching themes or issues. Additional tools, like the Pareto diagram, can incorporate stakeholder input and help prioritize problems to address.

Our first interventions centered on improving our QI processes in the laboratory. The main goal of these initiatives was to increase accountability and awareness of the scope and range of diagnostic errors, in a way that would have minimal impact on day-to-day clinical workflow. Most physicians and sonographers in our laboratory had voted to focus on these QI processes and felt positively about these changes. Thus, despite the increased attention on recent diagnostic errors, the initiatives were perceived as an opportunity for learning, improvement, and identifying future QI efforts.

In addition to QI process improvements being recognized as important by laboratory members, this category was also ideally suited for making sustainable changes. We built on existing processes, which allowed us to make improvements without significantly increasing time or effort demands. Sustainability is an important consideration for all QI interventions. A modest but sustainable intervention will have greater long-term impact than a more extensive, temporary intervention. It is critical to recognize both available resources and limitations when designing sustainable QI interventions.

### 4.2. Future Directions

As depicted in Figure 1, this process is inherently cyclical. As initiatives are implemented, we will continue to evaluate laboratory diagnostic accuracy as a measure of success. Countermeasures will be developed and tracked, to allow us to assess any unforeseen negative impacts. Over time, this cyclical process will be utilized repeatedly to develop additional initiatives with different focuses to meet the changing state of the echocardiography laboratory. Our next initiatives will focus on improving imaging processes, as determined by members of the laboratory.

### 4.3. Limitations

Our method for identifying diagnostic errors, systematic review of TTE diagnostic errors for surgical patients ≤ 5 years, may not be representative of all diagnostic errors that occur in our pediatric echocardiography laboratory. We specifically narrowed our review to this age group in an effort to identify errors that were preventable, rather than a limitation of suboptimal patient windows. We focused on preoperative TTEs as a potential way to identify diagnostic errors with the greatest clinical impact. We recognize that narrowing the scope of our systematic review will lead to the potential omission of important errors. There are numerous uncontrollable factors that may impact diagnostic error rate—disease complexity, clinical volume, and personnel changes are common examples—and these may have an equal or greater effect on diagnostic accuracy than concurrent QI initiatives. It is important to recognize the multifaceted causes of diagnostic errors when determining the success or failure of an initiative.

## 5. Conclusions

A stepwise and team-based method that incorporates input from faculty and sonographers allowed us to develop successful QI initiatives in our laboratory. This approach can serve as a model for a collaborative QI process in other echocardiography laboratories.

## Figures and Tables

**Figure 1 children-09-01845-f001:**
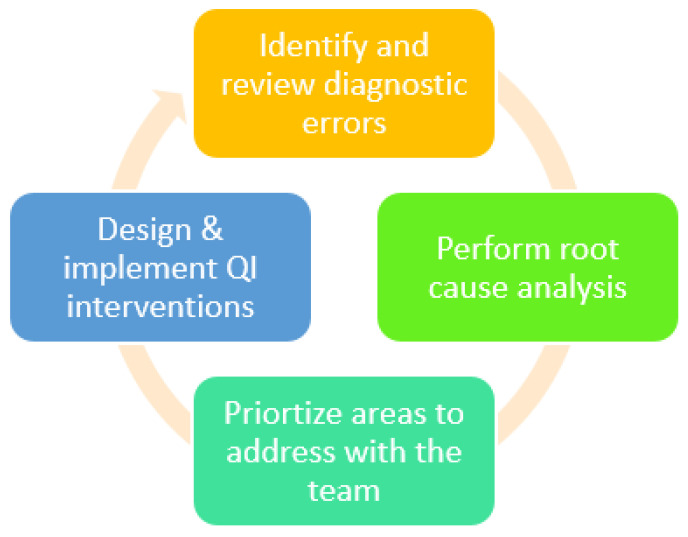
The quality improvement process. QI is both iterative and cyclical.

**Figure 2 children-09-01845-f002:**
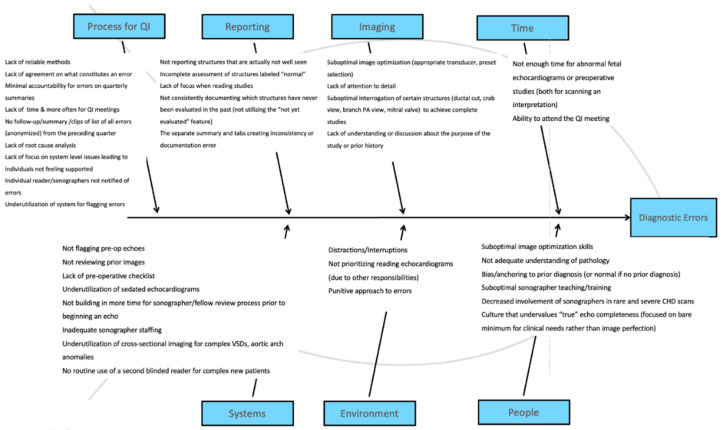
Fishbone diagram. This diagram depicts faculty and sonographer fellow responses, analyzed and grouped into seven categories that may contribute to diagnostic TTE errors.

**Figure 3 children-09-01845-f003:**
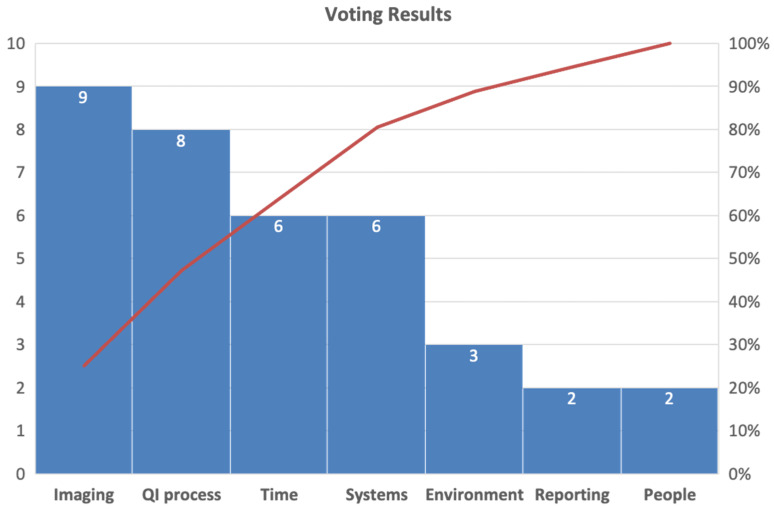
Pareto diagram depicting cardiology faculty and sonographer input on how to prioritize areas for QI intervention to address diagnostic TTE errors.

**Table 1 children-09-01845-t001:** Diagnostic errors identified in the baseline state.

Error	Anatomic Structure Involved	Type	Severity	Preventable
1	Ventricular septum	Discrepant diagnosis	Moderate	Preventable
2	Ventricular septum	False negative	Minor	Not preventable
3	Ventricular septum	Discrepant diagnosis	Moderate	Preventable
4	Atrial septum	Discrepant diagnosis	Minor	Preventable
5	Atrial septum	False negative	Minor	Preventable
6	Outflow tract	False negative	Severe	Preventable
7	Patent ductus arteriosus	Discrepant diagnosis	Minor	Preventable
8	Pulmonary arteries	False negative	Minor	Preventable
9	Atrial septum	False negative	Minor	Not preventable
10	Ventricular septum	False negative	Severe	Possibly preventable
11	Patent ductus arteriosus	False negative	Minor	Preventable
12	Mitral valve	Discrepant diagnosis	Minor	Not preventable
13	Atrial septum	False negative	Minor	Not preventable
14	Ventricular septum	Discrepant diagnosis	Minor	Preventable
15	Ventricular septum	False negative	Minor	Not preventable
16	Atrial septum	False negative	Minor	Preventable
17	Ventricles	Discrepant diagnosis	Minor	Preventable
18	Aortic arch	Discrepant diagnosis	Minor	Preventable

**Table 2 children-09-01845-t002:** Diagnostic errors identified in the follow up period.

Error	Anatomic Structure	Type	Severity	Preventable
1	Aortic arch	False negative	Minor	Preventable
2	Atrial septum	False negative	Minor	Preventable
3	Aortic arch	False negative	Minor	Possibly preventable
4	Inferior vena cava	False negative	Minor	Preventable
5	Atrial septum	False negative	Minor	Possibly preventable
6	Ventricular septum	Discrepant diagnosis	Minor	Preventable
7	Coronary arteries	Discrepant diagnosis	Minor	Possibly Preventable

## Data Availability

Not applicable.

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
