# Peer review of "Quality Improvement in a Pediatric Echocardiography Laboratory: A Collaborative Process"

_children, 2022, doi:10.3390/children9121845_

Round 1

Reviewer 1 Report

This paper is well written and the authors review basic methodology of QI but one important component of QI is sustainability of changes. It will be beneficial if they add discussion regarding how these changes are sustainable. 

Author Response

Point 1: This paper is well written and the authors review basic methodology of QI but one important component of QI is sustainability of changes. It will be beneficial if they add discussion regarding how these changes are sustainable.

Response: Thank you for these helpful comments. We agree sustainability is a key factor in quality improvement work and have added to our discussion to address this point. 

Reviewer 2 Report

Excellent presentation of QI processes within an echo lab from beginning to end. This report will provide an excellent example framework for echo labs looking to institute similar improvements. Appreciate the inclusion of fishbone diagram to further detail thought process behind your chosen interventions. 

Author Response

Point 1: Excellent presentation of QI processes within an echo lab from beginning to end. This report will provide an excellent example framework for echo labs looking to institute similar improvements. Appreciate the inclusion of fishbone diagram to further detail thought process behind your chosen interventions.

Response: Thank you for these kind comments. We indeed hope our project can provide a framework for other institutions who wish to initiate QI work or increase QI participation within pediatric echocardiographic laboratories (or other clinical settings).